# Mitochondrial Dynamics Markers and Related Signaling Molecules Are Important Regulators of Spermatozoa Number and Functionality

**DOI:** 10.3390/ijms22115693

**Published:** 2021-05-27

**Authors:** Isidora M. Starovlah, Sava M. Radovic Pletikosic, Tatjana S. Kostic, Silvana A. Andric

**Affiliations:** Laboratory for Reproductive Endocrinology and Signaling, Laboratory for Chronobiology and Aging, CeRES, DBE, Faculty of Sciences, University of Novi Sad, 21000 Novi Sad, Serbia; isidora.starovlah@dbe.uns.ac.rs (I.M.S.); sava.radovic@dbe.uns.ac.rs (S.M.R.P.); tatjana.kostic@dbe.uns.ac.rs (T.S.K.)

**Keywords:** acute/repeated psychological stress, mitochondrial dynamics and functionality markers, cAMP signaling markers, MAPK signaling markers, spermatozoa number and functionality

## Abstract

Here, we study possible mechanisms of (in/sub)fertility related to the acute or repeated psychological stresses (the most common stresses in human society) by following the transcriptional profile of 22 mitochondrial dynamics/function markers and 22 signaling molecules regulating both mitochondrial dynamics and spermatozoa number/functionality. An in vivo study mimicking acute (once for 3 h) and repeated (3 h for 10 consecutive days) psychophysical stress was performed on adult rats. The analysis of hormones, the number/functionality of spermatozoa, and 44 transcriptional markers were performed on individual samples from up to 12 animals per group. Results showed that both types of stress reduced spermatozoa functionality (acute by 4.4-fold, repeated by 3.3-fold) and ATP production (acute by 2.3-fold, repeated by 14.5-fold), while only repeated stress reduces the number of spermatozoa (1.9-fold). Stress significantly disturbed transcription of 34-out-of-44 markers (77%). Mitochondrial dynamics and functionality markers: 18-out-of-22 =>82% (mitochondrial-biogenesis-markers –>6-out-of-8 =>75%; mitochondrial-fusion-markers –>3-out-of-3 =>100%; mitochondrial-fission-markers –>1-out-of-2 =>50%; mitochondrial-autophagy-markers –>3-out-of-3 =>100%; mitochondrial-functionality-markers –>5-out-of-6 =>83%). Markers of signaling pathways regulating both mitochondrial dynamics/functionality and spermatozoa number/functionality important for male (in/sub)fertility –>16-out-of-22 =>73% (cAMP-signaling-markers –>8-out-of-12 =>67%; MAPK-signaling-markers –>8-out-of-10 =>80%). Accordingly, stress-triggered changes of transcriptional profile of mitochondrial dynamics/functionality markers as well as signaling molecules regulating both mitochondrial dynamics and spermatozoa number and functionality represent adaptive mechanisms.

## 1. Introduction

The miracle of life starts with fertilization and requires perfect spermatozoa functionality, which is a highly energy driven and demanding process regulated by complex as well as a wide network of signaling pathways. According to the World Health Organization (WHO), the overall burden of (sub/in)fertility is high, unknown in men, underestimated, and has not displayed any decrease over the last 20 years. Accordingly, the investigations of the mechanisms of (sub/in)fertility are urgent (https://www.who.int/r1eproductivehealth/topics/infertility/perspective/en/, accessed on 5 April 2021). Besides the aforementioned, many studies showed the increase of unexplained cases of infertile young males and decrease of the fertility rate in men younger than age 30. This is important, since the semen quality and male fertility are important as the fundamental marker of reproductive health and the fundamental biomarker of overall health [1,2]. The correlation between male (sub/in)fertility and stressful life has been reported [1,3,4]. The stress enhanced rat testicular germ cell apoptosis [5] and induced irreversible loss of germ cells and spermatozoa number [6]. Although stress is an adaptive response of an organism enabling survival and maintaining homeostasis [7], if repeated or persistent/chronic, it can cause diseases [8,9,10,11,12]. Different types of stressors, as well as major stressful life events, have been linked to reduced male reproductive function [3,13,14]. A higher number of stressful life events are observed in infertile men, and this was associated with a decline in semen quality during fertility treatment [3]. Epidemiological studies showed that DNA damage during stress-response is regulated through β2-adrenergic-receptors [15], strongly suggesting the importance of stress-signaling in regulation of mitochondrial homeostasis.

Mitochondria are key linking point between stress-response and spermatozoa functionality since they are responsible for satisfying enormous energy demands required for both processes [1,8,9,12]. Furthermore, both stress signaling and mitochondria are essential for spermatozoa functionality. The spermatogenesis and fertility are disturbed in α1-adrenergic-receptors-knockout-male-mice [16]. The functionality of mitochondria differentiates human spermatozoa with high and low fertilizing capability [17]. It has been shown that mtDNA depletion may play an important role in the pathophysiology of male infertility [18] serving as a diagnostic marker of sperm quality in infertile men [19]. Additionally, the stages of spermatogenesis are characterized with the changes in mitochondrial morphology [20], and these changes shed light on the unexplained asthenozoospermia with altered ultrastructure of mitochondria [21]. Defects of mtDNA in oligoasthenozoospermic patients made DNA unavailable for amplification [22]. Large-scale deletions of mtDNA were pointed to as risk factors for poor sperm quality in asthenoteratozoospermia-induced male infertility [23]. Beside mtDNA, the importance of mitochondrial membrane potential is recognized not only for spermatozoa functionality [24,25,26], but also, in combination with sperm DNA fragmentation, as standard semen parameter for the prediction of natural conception [27]. Moreover, mitochondrial transcription factor TFAM (Transcription Factor A, Mitochondrial) causes the reduction of mtDNA content in human sperm [28], and the expression of TFAM gene positively correlates with abnormal forms, sperm DNA fragmentation, and mtDNA copy number [29,30]. Human sperm motility and viability are regulated by mitophagy [31]. The loss of human spermatozoa motility is mitigated by UCP2 (uncoupling protein 2) [32], while the motility and cryoprotective potential of human sperm is connected with the MFN2 (mitofusin 2) expression level [33]. According to everything mentioned above, the mitochondria are a crucial organelle for spermatozoa functionality, and their homeostasis strongly correlates with (sub/in)fertility [1]. Although profiles of signaling proteins in human spermatozoa suggested that the phosphorylated levels of several proteins were significantly associated with motility parameters [34] and these proteins regulate mitochondrial-network-homeostasis, to the best of our knowledge, there are no published pieces of evidence about the transcriptional profile of mitochondrial dynamic markers and related signaling molecules in spermatozoa from a stressed organism.

The mitochondrial network homeostasis is kept and maintained by well-coordinated, but intriguing, processes of mitochondrial dynamics including the complex mitochondrial protein-import machinery (mitochondrial transduceom), movement of mitochondria to position themselves strategically in the cell (motility/trafficking), mitochondrial biogenesis, mitofusion, mitofission, and mitophagy [35,36,37,38]. All of these are interplay of a sophisticated and multistep molecular events required for renewal, adaptation, or expansion of mitochondrial network in a cell during episodes of damage or periods of intensified energy demand [28,39,40,41]. Nucleo-mitochondrial interactions are required for the spatiotemporal regulation of mitochondrial dynamics and are dependent on the interplay between transcription factors and members of the PGC1 (peroxisome proliferator-activated receptor gamma coactivator 1) family of coactivators (PGC1α, PGC1β, PRC) regulating the expression of main markers of mitochondrial dynamic [35,36]. These main markers of mitochondrial dynamics include markers of mitochondrial biogenesis (PGC1α, PGC1β, NRF1, NRF2 (nuclear respiratory factor 1 and 2), TFAM), mitofusion (MFN1, MFN2, OPA1 (mitochondrial dynamin like GTPase)), mitofission (DRP1 (dynamin-related protein 1), FIS1 (mitochondrial fission protein 1)), and mitophagy (PINK (PTEN induced kinase), PARKIN), as well as important markers of the respiratory chain function [35,36,37,38,40,41]. Mitochondrial dynamic processes are controlled by an intriguing and complex network of cellular signaling pathways [36,39,41] conveying different environmental signals: stress [42,43], temperature [44], energy deprivation [41], availability of nutrients [41], and growth factors [45]. More importantly, all signaling pathways regulating mitochondrial dynamics are essentially involved in the regulation of spermatozoa function. The recent review article pointed out the great gap and interest in the research related to mitochondria and male fertility [46]. Many studies discussed the correlation between stress and/or stressful life and male (in)fertility [3]. Yet, the mechanisms are not described.

Since our previous ex vivo studies showed importance of adrenergic signaling in regulation of mitochondrial dynamics markers, here we hypothesize that the psychophysical stress alters the changes of mitochondrial dynamics and functionality markers as well as signaling molecules regulating both mitochondrial dynamics and spermatozoa number and functionality.

## 2. Results

In the search for any possible mechanism(s) of the (in/sub)fertility related to spermatozoa functionality, the in vivo approach was designed to mimic situations in human population exposed to acute and repeated psychological stress, the most common stress in human society. This was achieved by applying the immobilization (IMO) stress of 3 h for once (1×3hIMO, acute) or for 10 consecutive days (10×3hIMO, repeated) on the adult male rat [13,14,47].

### 2.1. Repeated Psychophysical Stress Increases the Level of Stress Hormones in Circulation, but Decreases Androgens Levels, as Well as the Functionality, ATP Level and Number of Spermatozoa

The effects of the acute (1×3hIMO) as well as repeated (10×3hIMO) stress were in agreement with previous studies [13,14,47] since IMO was effective as a stressor (Figure 1) elevating the serum adrenaline of acutely (1×3hIMO –>3.1-fold) and repeatedly (10×3hIMO –>2.9-fold) stressed animals. Similar effects were observed on the level of corticosterone (1×3hIMO –>6.6-fold; 10×3hIMO –>6.4-fold =>544.1%). Conversely, circulating androgens (T + DHT) were reduced in both stressed groups (1×3hIMO –>5.7-fold, 10×3hIMO –>10.5-fold). Repeated stress (10×3hIMO) significantly reduced (1.9-fold) the number of spermatozoa, while both types of stress significantly inhibited spermatozoa functionality (1×3hIMO –>4.4-fold; 10×3hIMO –>3.3-fold) as well as ATP production (1×3hIMO –>2.3-fold, 10×3hIMO –>14.5-fold).

Unlike ever before, the transcriptional profile of mitochondrial dynamics markers and signaling molecules regulating both mitochondrial dynamics and spermatozoa number and functionality (important for fertilization) were explored to reveal any possible mechanism(s) beyond these effects. Results showed that stress dramatically disturbed the expression of transcripts for the markers of mitochondrial dynamics and functionality as well as related signaling pathways, since the expression levels of 34-out-of-44 (70%) were changed (Figure 2, Figure 3 and Figure 4).

### 2.2. Transcriptional Profiles of Mitochondrial Dynamics and Functionality Markers Are Dramatically Changed in Spermatozoa from Stressed Rats

Markers of mitochondrial dynamics and functionality in spermatozoa are under significant influence of stress hormones since transcriptional levels of 18-out-of-22 (82%) were changed (Figure 2).

Levels of 6-out-of-8 (75%) transcripts for mitochondrial biogenesis markers increased in spermatozoa in stressed rats. Acute stress (1×3hIMO) significantly increased spermatozoal *Ppard* (1.9-fold), while effects of repeated stress (10×3hIMO) were more prominent since transcription of 6-out-of-8 =>75% markers were increased. The level of *Ppargc1a*, encoding PGC1, the master regulator involved in transcriptional control of all processes related to mitochondrial homeostasis and integrator of environmental signals [35,36], significantly increased (2.7-fold). This was followed by increased transcription of its down-stream-targets that act on genes for subunits of the oxidative phosphorylation (OXPHOS): *Nrf1* (2.3-fold), *Nrf2a* (1.8-fold), *Tfam* (1.9-fold), *mtNd1* (7-fold), and *Ppard* (1.8-fold).

Levels of transcripts for all (3-out-of-3 =>100%) mitochondrial fusion markers increased in spermatozoa in repeatedly stressed rats. Repeated stress significantly increased all spermatozoal mitofusion and mito-architecture markers: *Mfn1* (3.3-fold), *Mfn2* (2.9-fold) and *Opa1* (1.6-fold).

Level of transcript for 1-out-of-2 (50%) mitochondrial fission marker increased in spermatozoa in repeatedly stressed rats. In the same samples used to obtain the aforementioned results, repeated stress increased *Drp1* (3.2-fold), while *Fis1* remained unchanged.

Levels of transcripts for all (3-out-of-3 =>100%) mitochondrial autophagy markers increased in spermatozoa in repeatedly stressed rats. Repeated stress increased transcription of *Pink1* (2.3-fold), *Prkn* (1.45-fold), and *Tfeb* (2.6-fold).

Levels of transcripts for 5-out-of-6 (83%) mitochondrial functionality markers were changed. In the same samples used to obtain the aforementioned results, significant changes in the transcriptional profiles of mitochondrial functional markers as well as NRF1/NRF2 downstream targets (CytC, COX4 and UCPs) were observed in spermatozoa in repeatedly stressed rats. Increased expressions of transcripts for *Cox4i1* (2.7-fold), *Cox4i2* (2.8-fold) and *CytC* (1.6-fold) were detected. Furthermore, repeated stress significantly changed the transcriptional profile of genes encoding proteins regulating the proton leak and the production of reactive oxygen species [44]. The increased level of transcript for *Ucp2* (3.2-fold), the most prominently expressed UCP gene in spermatozoa (*Ucp2*-Ct = 22.65 > *Ucp3*-Ct = 29.24 > *Ucp1*-Ct = 29.73), was observed in repeatedly stressed rats spermatozoa. Conversely, in the same samples, the level of transcript for *Ucp3* decreased (5.1-fold), while *Ucp1* remained unchanged.

Given the central importance of cAMP and MAPK signaling, not only for regulation of mitochondrial dynamics and functionality [35,36,40], but also for regulation of spermatozoa number and functionality [34], transcriptional profiles of main signaling molecules were explored.

### 2.3. Transcriptional Profiles of Signaling Molecules Regulating Mitochondrial Dynamics and Functionality as Well as Number and Functionality of Spermatozoa, All of Which Are Important for Fertility, Are Dramatically Changed in Spermatozoa from Stressed Rats

The markers of signaling pathways regulating both mitochondrial dynamics and functionality as well as spermatozoa number/functionality important for male (in)fertility are under significant influence of stress hormones, since transcriptional levels of 16-out-of-22 (73%) were changed (Figure 3).

Levels of transcripts for 8-out-of-12 (67%) cAMP signaling-markers were changed in spermatozoa in stressed rats. Acute stress (1×3hIMO) significantly decreased spermatozoal transcripts for *Adcy6* (1.9-fold), and *Adcy8* (2.3-fold). Repeated stress (10×3hIMO) exhibited the opposite effect on *Adcy6* (3.1-fold increase), yet a similar effect on *Adcy8* (2.3-fold decrease). In the same spermatozoa, the level of transcript for *Adcy9* increased (2.7-fold). In parallel, repeated stress significantly increased the expression of transcripts for genes encoding catalytic and regulatory PRKA subunits: *Prkaca* (1.7-fold), *Prkacb* (2.4-fold), *Prkar1a* (2.3-fold), *Prkar2a* (1.9-fold), and *Prkar2b* (3.9-fold).

Levels of transcripts for 8-out-of-10 (80%) MAPK signaling markers increased in spermatozoa in stressed rats. Acute stress significantly increased spermatozoal transcripts for *Mapk1* (1.9-fold), *Mapk6* (1.7-fold), and *Mapk8* (1.6-fold). Repeated stress significantly increased levels of transcripts for *Mapk1* (3.3-fold), *Mapk3* (1.7-fold), *Mapk6* (1.6 -fold), *Mapk8* (3.2-fold), *Mapk9* (1.8-fold), *Mapk12* (2.4-fold), *Mapk13* (3.2-fold), and *Mapk14* (1.8-fold).

## 3. Discussion

For the first time our results show that stress triggers changes of transcriptional profile of mitochondrial dynamics and functionality markers as well as signaling molecules (Figure 4) regulating both mitochondrial dynamics and spermatozoa number and functionality. Since most of the effects (91%) on the level of the transcripts are a significant increase (please see Table 1) they could represent adaptive mechanisms to keep spermatozoa functionality. With changing the transcriptional profiles, spermatozoa may be trying to preserve the basic mitochondrial network homeostasis and self-activity since both types of stress significantly decreased ATP production. It is important to point out that affected molecules are essential for spermatozoa functionality and accordingly also for (in/sub)fertility. Several lines of evidence prove the above mentioned. (1) Repeated psychophysical stress reduces the number of spermatozoa. (2) Both types of stress, acute and repeated, significantly reduced spermatozoa functionality and ATP level in spermatozoa. (3) Stress significantly disturbed transcription of 77% markers of mitochondrial dynamics/functionality as well as signaling pathways regulating both mitochondrial homeostasis and fertility. All aforementioned signaling molecules are very well known as essential regulators of spermatozoa number and functionality [34], as well as regulators of mitochondrial dynamics markers [35,36,40], and therefore are very important for (in/sub)fertility.

Results presented in this and our recently published article [14] are in line with others showing that chronic intermittent stress decreases sperm number [6], the number of spermatogenic cells [48], sperm motility [49], and sperm quality [50] in male rats. The inhibitory role of stress hormones on sperm functionality is in line with stress-induced-GRs-signaling-mediated spermatogenesis impairment [51] as well as reduced testosterone and sperm motility in high and moderate male runners [52]. Furthermore, fertility and spermatogenesis are altered in α1-ADRs-knockout-male-mice [16]. In humans, stress was associated with a reduction in percentage of progressively motile sperm [53] and patients with post-traumatic stress disorder have higher secondary infertility [54]. All of the aforementioned suggests the important and complex involvement of stress signaling in spermatogenesis and fertility.

It is clear that the repeated stress dramatically changed transcriptional pattern of mitochondrial dynamics/functionality markers as well as signaling molecules regulating both mitochondrial homeostasis and fertility by increasing the transcription of 30-out-of-32 (94%) genes (Table 1 and Figure 4). This increase could be the possible adaptive mechanism to keep energy balance and preserve capabilities for fertilization. It is difficult to compare our finding since there is not a lot of evidence available from other authors. However, it has been shown that *PPARGC1A* and *PRKAR1A* are changed in spermatozoa of type 2 diabetes mellitus patients [55]. Furthermore, an increase in *Nrf2* diminished testicular inflammation [56], while TFAM gene expression positively correlates with abnormal forms, sperm DNA fragmentation, and mtDNA copy number [29,30]. TFAM is essential for mammalian mtDNA transcription in animals and humans. It has been reported that a testis-specific mouse TFAM isoform lacks the mitochondrial targeting sequence and is present only in the nucleus of spermatocytes and elongating spermatids [57]. Our observations about significant increase in the level of *Ucp2* in spermatozoa from both stressed and adrenaline-stimulated spermatozoa [14] could be a possible explanation for findings of other authors showing that the presence of UCP2 mitigates the loss of human spermatozoa motility [32]. Since the transcripts for main markers (*Mfn1*, *Mfn2*, *Opa1*) of mitochondrial fusion/architectures, important for homeostasis of mitochondrial network and functionality [35,36,40], were dramatically increased in spermatozoa from stressed rats, these results may explain results of other authors showing the positive relation of the expression level of MFN2 with motility and cryoprotective potential of human sperm [33], as well as mitofusin-mediated promotion of OXPHOS [58]. It has been shown that *Cox4i1* (gene encoding the terminal enzyme in the mitochondrial respiratory chain and transcript increased in spermatozoa from stressed rats) is also significantly increased in sperm of obese males mice [59] and could be important for male infertility treatment [60]. Additionally, our results showing that repeated stress increased *Cytc* is related with the findings that perturbed mitochondrial release of CytC in human spermatozoa may be the distinguishing molecular feature between oligospermia and asthenospermia [61].

The final important insight from our study is that the psychophysical stress significantly disturbed transcription of signaling pathways (cAMP and MAPK signaling) regulating both mitochondrial dynamics and spermatozoa number/functionality. All aforementioned signaling molecules are very well known as essential regulators of spermatozoa number and functionality [34], as well as regulators of PGC1, the biogenesis of OXPHOS, mitofusion, mitofission, and mitophagy [35,36,40] and that they are very important for fertility. It is very difficult to provide precise mechanism since it is very well known that signaling network in spermatozoa is complex and very precisely regulated to provide fertility homeostasis in health and diseases [62]. Besides, there is not many published evidence. However, our results showing increased transcripts for most of adenylyl cyclases (ADCY), as well as subunits of PRKA are in line with other authors who showed that adenosine and catecholamine analogs activate sperm motility by mechanisms that require atypical sperm ADCY and PRKACA [63] and that cAMP pathways are compartmentalized in sperm, with ADCY1-9 in the head and ADCY10 and PRKA in the flagellum [64]. Furthermore, PRKAR2A reduction in asthenozoospermic will likely decrease sperm quality [65], while *Prkar2b* is a potential heat sensitive target in germ cells [66]. Everything mentioned above is very important because CatSper channels are regulated by PRKA [67]. Additionally, our recent publication showed that ex vivo manipulation of stress signaling in spermatozoa changed transcriptional profiles of 84% mitochondrial dynamics/functionality markers and that most of these effects are mediated through α1-and/or-β-adrenergic receptors [14], which are very well known activators of ADCY-cAMP-PRKA-signaling [15]. In relation to MAPK signaling, our results are showing increased transcription of 80% of MAPKs, and they could be compared with findings that testicular hyperthermia induces both MAPK1/3 and MAPK14 [68] and that MEK1/2 and ERK2 regulate actin polymerization associated with spermatozoa capacitation [69].

Why is all of the aforementioned important? It has been shown that the stressful life, i.e., the alpha males (“life at the top”) exhibited much higher stress hormone levels than second-ranking (beta) males [70]. Furthermore, spermatozoa functionality and male fertility are important not only as the fundamental markers of reproductive health, but also as the fundamental biomarker of overall health. Accordingly, urgent reactions are essential [2,3]. Why do we believe that our results provide a completely new view on testing of spermatozoa functionality and (sub/in)fertility? To prove that these results have direct and strong translational significance, we started analysis on human subjects. Preliminary results (Starovlah/Tomanic et al., unpublished results) showed correlation between different transcriptional profiles of mitochondrial dynamics markers and different types of spermiograms. Moreover, progesterone (acrosome-reaction-inducer) changed the transcriptional profile of some mitochondrial dynamics markers, suggesting direct relation with in vivo events required for fertilization. Accordingly, all presented molecular markers can be used as “mitochondrial-sperm-signature” to test spermatozoa functionality and (sub/in)fertility. These results are giving a new understanding of the correlation between stress and other life-style-environmental-one-health-factors, and male (in/sub)fertility. We believe that our work will be of significant interest to both basic research and clinical practice. This is because the preliminary results on a small number of human samples (12) showed dramatic changes in transcriptional profiles of mitochondrial dynamics markers between spermatozoa of teratozoospermic, asthenoteratozoospermic, and oligoasthenoteratozoospermic patients and normozoospermic patients. Moreover, according to the questionnaire completed by 89 patients, 91% reported some degree of stress: 44%—low degree of stress, 36%—frequent stressful situations, 11%—high degree of stress (Starovlah/Tomanic et al. unpublished results).

## 4. Materials and Methods

All experiments were carried out in the Laboratory for Reproductive Endocrinology and Signaling and Laboratory for Chronobiology and Aging (wwwold.dbe.pmf.uns.ac.rs/en/nauka-eng/lares, accessed on 5 April 2021). The methods used in this study were carried out in accordance with the relevant guidelines and regulations. These methods were previously reported by our group (for all references please see [13,14,46]) and are outlined briefly here, but more details are available in Appendix A.

### 4.1. Statement of Institutional Review Board

The manuscript is approved by the Committee of the Faculty of Sciences, University of Novi Sad.

### 4.2. Statement That the Authors Complied with ARRIVE Guidelines and Institutional Animal Care and Use Committee Guidelines

The authors complied with ARRIVE guidelines and all experiments were in adherence to the ARRIVE guidelines. Besides, all experimental protocols were approved (statement no. 01-201/3) by the local Ethical Committee on Animal Care and Use of the University of Novi Sad operating under the rules of National Council for Animal Welfare and the National Law for Animal Welfare (copyright March 2009), following the NRC publication Guide for the Care and Use of Laboratory Animals and NIH Guide for the Care and Use of Laboratory Animals.

### 4.3. Animals and Experimental Model of Stress

Three-months-old male *Wistar* rats were used for the experiments. Model of psychophysical stress by immobilization (IMO) was performed in the morning by the method previously described [13,14,46]. Briefly, animals were divided into the following groups: control—unstressed rats; 1×3hIMO—rats subjected to IMO once, for 3 h; 10×3hIMO—rats subjected to repeat IMO of 3 h for 10 consecutive days. At the end of IMO period, animals were quickly decapitated without anesthesia. Serum samples were collected and assayed for androgens (testosterone + dihydrotestosterone; T + DHT), adrenaline, and corticosterone (CORT) levels. The experiment was repeated three times.

### 4.4. Serum Hormones Measurement

Measurement of hormone levels in serum was duplicated. The level of androgens was referred to as T + DHT since the anti-testosterone serum №250 showed 100% cross-reactivity with DHT (sensitivity: 6 pg per tube; intra-assay coefficient of variation 5–8%). Adrenaline levels were measured using the adrenaline research ELISA Kit (www.ldn.de, accessed on 5 April 2021) with the standard range of 0.45–45 ng/mL and detection limit of 3.9 pg/mL. Corticosterone (CORT) levels were measured by the corticosterone EIA Kit (www.caymanchem.com, accessed on 5 April 2021) with 30 pg/mL as the lowest standard significantly different from blank.

### 4.5. Isolation of Spermatozoa and Assessment of Their Functionality (Capacitation and Acrosome Reaction)

Spermatozoa were isolated from caudal epididymides following the WHO laboratory manual (www.who.int/reproductivehealth/publications/infertility/9789241547789/en/, accessed on 5 April 2021) with modifications for rat spermatozoa isolation (for reference please see [14]). Caudal epididymides were quickly removed, placed in a Petri dish containing the medium for isolation and preservation of spermatozoa, finely punctuated with a needle, and incubated at 37 °C for 10 min. Released spermatozoa were collected, centrifuged for 5 min at 700× *g* and resuspended in the appropriate medium. Concentrations of isolated spermatozoa were calculated using a Makler counting chamber. *To determine the functionality of the spermatozoa*, approximately 1.5 × 10^5^ spermatozoa in Whitten’s Media were mixed with WH+ media (Whitten’s Media supplemented with the 10 mg/mL BSA and 20 mM of NaHCO_3_) with a drop of mineral oil, for 1 h at 37 °C. Capacitated spermatozoa were incubated without or with 15 μM progesterone (PROG) for the activation of the acrosome reaction, for 30 min at 37 °C. After the incubation, spermatozoa suspension was fixed for 20 min and centrifuged for 1 min at 12,000× *g*. Spermatozoa in pellet were washed with 100 mM ammonium acetate. The smears of fixed spermatozoa were air-dried and stained by covering the slides with staining solution, containing 0.04% Coomassie Blue—G250, for 5 min at room temperature, rinsed with distilled water and allowed to air-dry. The stained smears were analyzed using the microscope Leica DMLB 100 T with 1000× magnification. Ten to fifteen photos per slide were taken by Leica MC190 camera and LAS Version 4.8.0 software and up to 100 spermatozoa per slide counted to determine the acrosome status. Blue staining in the acrosome region of the head indicated intact acrosome, whereas spermatozoa without blue staining in the acrosome region were considered acrosome-reacted. Data are presented as the percentage of acrosome-reacted spermatozoa ± SEM.

### 4.6. Measurement of ATP Level

The ATP level was determined using ATP Bioluminescence CLS II kit following the manufacturer’s instructions (www.sigmaaldrich.com, accessed on 5 April 2021) Spermatozoa (1 × 10^6^) were lysed in distilled H_2_O and mixed with Tris-EDTA in ratio of 1:9 (*v*/*v*). Lysed spermatozoa in Tris-EDTA were incubated in water bath (100 °C/3 min), centrifuged for 1 min at 900× *g*. Supernatant was used for the measurement of ATP while the pellet was further used for measurement of protein concentration using Bradford protein assay. Sample and standard were mixed with Luciferase reagent in ratio of 1:1 (*v*/*v*), and luminescence was measured by the Biosystems/luminometer (Fluoroscan, Ascent, FL, ThermoLabsystems, Helsinki, Finland), as described previously by our group (for references please see [71]).

### 4.7. RNA Isolation and cDNA Synthesis

Isolation of total RNA was done using GenElute™ Mammalian Total RNA Miniprep Kit (www.sigmaaldrich.com, accessed on 5 April 2021), following the DNase I (RNase-free) treatment (www.neb.com, accessed on 5 April 2021). cDNA was synthesized using the High Capacity Kit (www.thermofisher.com, accessed on 5 April 2021). RNA quality and DNA integrity were checked using control primers for *Gapdh*.

### 4.8. Relative Quantification of Gene Expression

Relative quantification of gene expression was done by real-time PCR (RQ-PCR) using SYBR^®^Green-based chemistry (www.thermofisher.com, accessed on 5 April 2021) in the presence of cDNA and specific primers (Appendix A). The transcription of *Gapdh* was used to correct the variations in cDNA content between samples. Relative quantification of each gene was performed in duplicate, three times for each sample of three independent experiments.

### 4.9. Statistical Analysis

Results of the experiments represent group means ± SEM values of the individual variation from three independent experiments (3 to 6 rats per group). Results from each experiment were analyzed by Mann–Whitney’s unpaired non-parametric two-tailed test (for two-point data experiments), or by one-way ANOVA for group comparison, followed by Student–Newman–Keuls multiple range test. All the statistical analysis was done using GraphPad Prism 5 Software (GraphPad Software 287 Inc., La Jolla, CA, USA). In all cases, *p*-value < 0.05 was considered to be statistically significant.

## 5. Conclusions

Psychophysical stress disturbs signaling pathways and molecules responsible for mitochondrial dynamics and functionality in spermatozoa with consequences for their functionality. Repeated stress changes 18-out-of-22 mitochondrial dynamics and functionality markers as well as 16-out-of-22 signaling molecules regulating mitochondrial dynamics/functionality in spermatozoa leading to reduced spermatozoa number/functionality important for male (in/sub)fertility. Stress-triggered changes of transcriptional profile of mitochondrial dynamics and functionality markers as well as signaling molecules regulating both mitochondrial dynamics and spermatozoa number and functionality represent adaptive mechanisms to keep spermatozoa functionality since ATP level is low. They do not only correlate with, but are also essential for spermatozoa functionality and (in/sub)fertility, considering that all events depend on the same regulators.

## Figures and Tables

**Figure 1 ijms-22-05693-f001:**
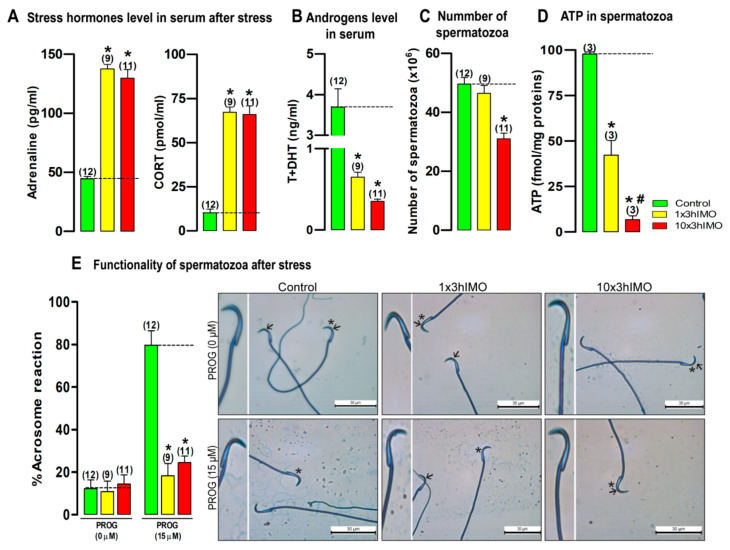
The repeated psychophysical stress increases the level of stress hormones in circulation, but decreases androgens levels, as well as the functionality, ATP level, and number of spermatozoa. The circulating (serum) levels of (**A**) stress hormones adrenaline and corticosterone (CORT), as well as (**B**) androgens (testosterone + dihydrotestosterone, T + DHT) after psychophysical stress by immobilization. (**C**) The number of spermatozoa isolated from caudal epididymides of unstressed rats (control), rats subjected to acute immobilization (IMO) stress once, for 3 h (1×3hIMO), and rats subjected to repeat IMO of 3 h for 10 consecutive days (10×3hIMO). (**D**) ATP content in spermatozoa isolated from control, acutely (1×3hIMO) or repeatedly (10×3hIMO) stressed rats. (**E**) The functionality of spermatozoa (% of acrosome reacted spermatozoa) isolated from control and acutely (1×3hIMO) and repeatedly (10×3hIMO) stressed rats. Capacitated spermatozoa were stimulated with progesterone (PROG 15 µM) in parallel with spermatozoa not treated with progesterone (PROG 0 µM). Blue staining in the acrosome region of the head indicated intact acrosome, whereas spermatozoa without blue staining in the acrosome region were considered to be acrosome reacted. Arrows indicate acrosome intact spermatozoa; scale bar 30 µm. Star indicates spermatozoa magnified on the right panel. Data bars are mean ± SEM values of three independent in vivo experiments. Statistical significance was set at level *p* < 0.05: * vs. control group; # vs. 1×3hIMO group.

**Figure 2 ijms-22-05693-f002:**
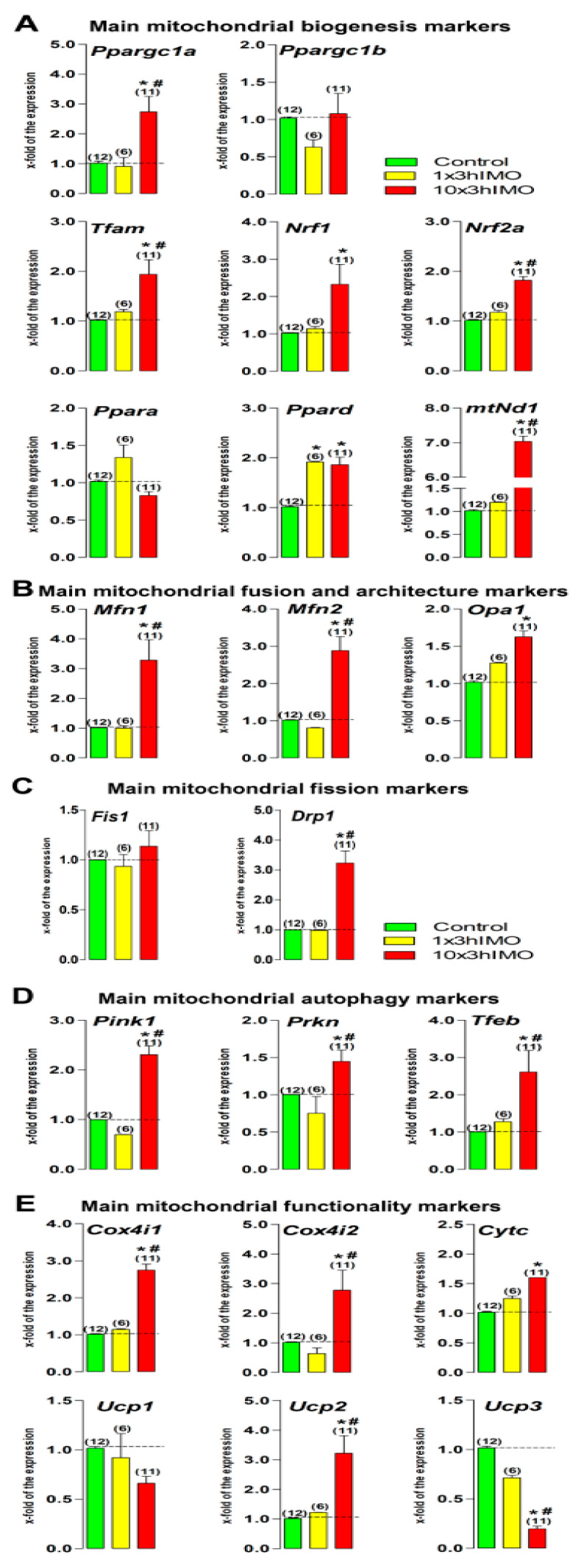
The transcriptional profiles of mitochondrial dynamics and functionality markers are significantly changed in spermatozoa of repeatedly stressed adult rats. RNA isolated from spermatozoa obtained from undisturbed and stressed rats were used for analysis of the transcriptional profile of markers of mitochondrial biogenesis (**A**), mitochondrial fusion and architecture (**B**), mitochondrial fission (**C**), mitochondrial autophagy (**D**), and mitochondrial functionality (**E**). Data bars are mean ± SEM values of three independent in vivo experiments. Statistical significance was set at level *p* < 0.05: * vs. control group; # vs. 1×3hIMO group.

**Figure 3 ijms-22-05693-f003:**
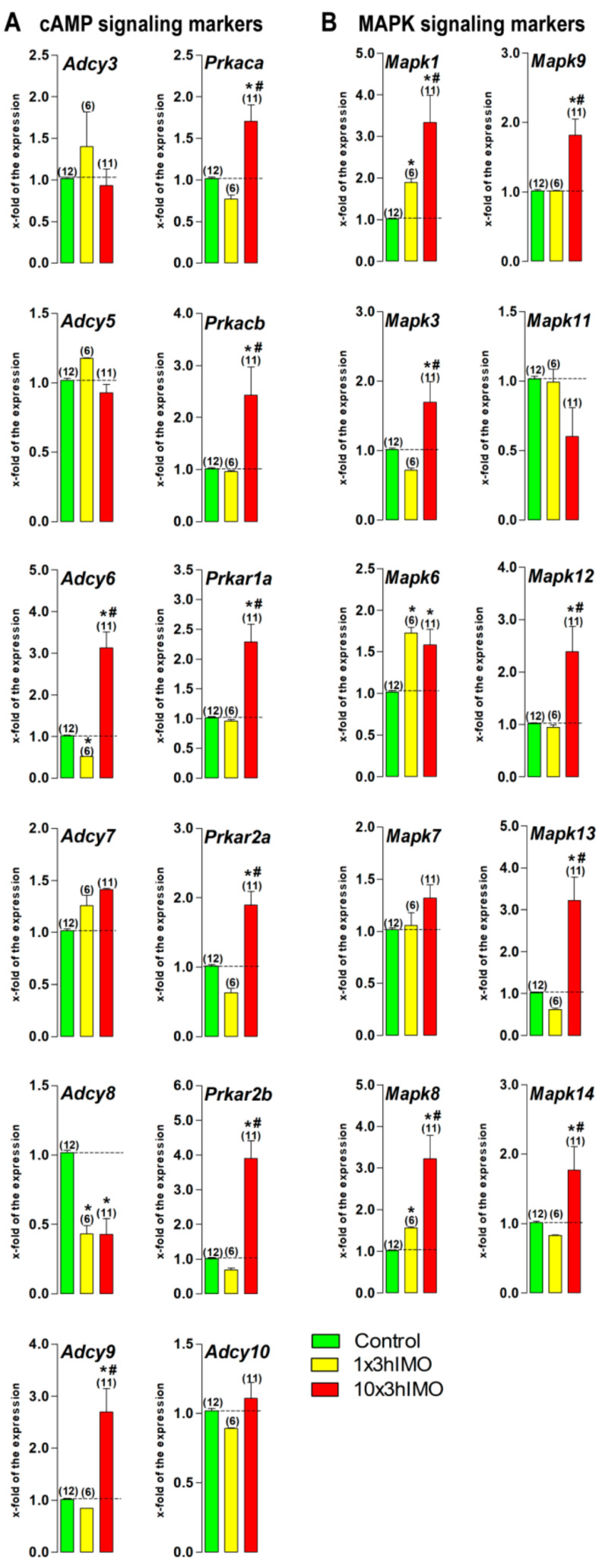
The transcriptional profiles of signaling molecules regulating mitochondrial dynamics and functionality as well as spermatozoa number and functionality are changed in spermatozoa of repeatedly stressed adult rats. RNA isolated from spermatozoa obtained from undisturbed and stressed rats were used for analysis of the transcriptional profile of markers of cAMP signaling (**A**) and MAPK signaling (**B**) pathways. Data bars are mean ± SEM values of three independent in vivo experiments. Statistical significance was set at level *p* < 0.05: * vs. control group; # vs. 1×3hIMO group.

**Figure 4 ijms-22-05693-f004:**
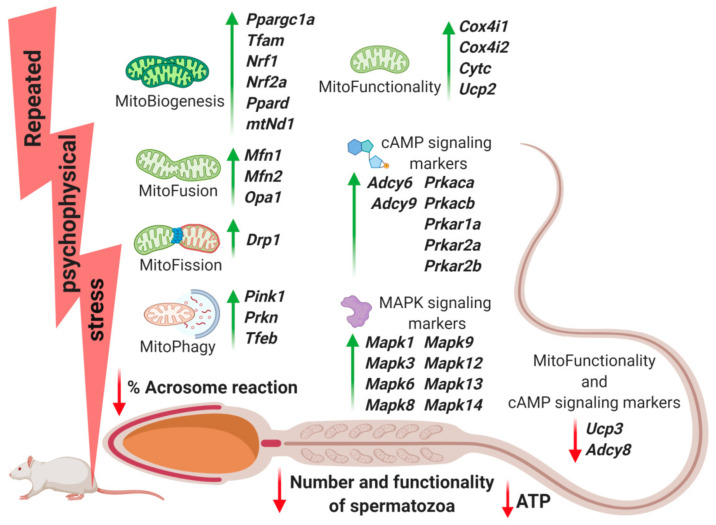
Repeated psychophysical stress disturbs the transcriptional profiles of mitochondrial dynamics and functionality markers and related signaling molecules, leading to the decrease in spermatozoa number and functionality. Repeated stress reduces the number of spermatozoa (1.9-fold =>48%), spermatozoa functionality (3.3-fold =>69%), and ATP levels (14.5-fold =>93%), as well as significantly disturbed transcription of 77% (34-out-of-44) markers. Mitochondrial dynamics and functionality –>18-out-of-22 =>82% (mitochondrial-biogenesis-markers –>6-out-of-8 =>75%; mitochondrial-fusion-markers –>3-out-of-3 =>100%; mitochondrial-fission-markers –>1-out-of-2 =>50%; mitochondrial-autophagy-markers –>3-out-of-3 =>100%; mitochondrial-functionality-markers –>5-out-of-6 =>83%). Markers of signaling pathways regulating both mitochondrial dynamics/functionality and spermatozoa number/functionality important for male (in/sub)fertility –>16-out-of-22 =>73% (cAMP-signaling-markers –>8-out-of-12 =>67%; MAPK-signaling-markers –>8-out-of-10 =>80%).

**Table 1 ijms-22-05693-t001:** The transcriptional profiles of mitochondrial dynamics and functionality markers and signaling molecules regulating mitochondrial dynamics and functionality as well as spermatozoa number and functionality are significantly changed in spermatozoa of repeatedly stressed adult rats. Data are presented as means ± SEM values of three independent experiments. Statistical significance at level *p* < 0.05: * vs. control group; # vs. 1×3hIMO group.

	Group	Control	1×3hIMO	10×3hIMO
Transcript	
***Ppargc1a***	**1.0** ± 0.07	**0.91** ± 0.30	**2.74 ***^,#^ ± 0.52 
***Tfam***	**1.0** ± 0.02	**1.19** ± 0.04	**1.94 ***^,#^ ± 0.29 
***Nrf1***	**1.0** ± 0.02	**1.14** ± 0.06	**2.33 *** ± 0.54 
***Nrf2a***	**1.0** ± 0.06	**1.18** ± 0.04	**1.82 ***^,#^ ± 0.08 
***Ppard***	**1.0** ± 0.01	**1.92 *** ± 0.01 	**1.86 *** ± 0.15 
***mtNd1***	**1.0** ± 0.02	**1.20** ± 0.01	**7.03 ***^,#^ ± 0.16 
***Mfn1***	**1.0** ± 0.01	**1.00** ± 0.08	**3.28 ***^,#^ ± 0.69 
***Mfn2***	**1.0** ± 0.04	**0.81** ± 0.01	**2.88 ***^,#^ ± 0.37 
***Opa1***	**1.0** ± 0.06	**1.27** ± 0.01	**1.63 *** ± 0.08 
***Drp1***	**1.0** ± 0.01	**0.98** ± 0.03	**3.23 ***^,#^ ± 0.39 
***Pink1***	**1.0** ± 0.00	**0.70** ± 0.01	**2.31 ***^,#^ ± 0.18 
***Prkn***	**1.0** ± 0.05	**0.75** ± 0.23	**1.45 ***^,#^ ± 0.15 
***Tfeb***	**1.0** ± 0.01	**1.28** ± 0.08	**2.61 ***^,#^ ± 0.59 
***Cox4i1***	**1.0** ± 0.02	**1.15** ± 0.02	**2.75 ***^,#^ ± 0.17 
***Cox4i2***	**1.0** ± 0.02	**0.64** ± 0.19	**2.78 ***^,#^ ± 0.68 
***Cytc***	**1.0** ± 0.01	**1.25** ± 0.04	**1.60 *** ± 0.08 
***Ucp2***	**1.0** ± 0.02	**1.23** ± 0.01	**3.22 ***^,#^ ± 0.59 
***Ucp3***	**1.0** ± 0.06	**0.71** ± 0.03	**0.20 ***^,#^ ± 0.02 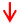
***Adcy6***	**1.0** ± 0.02	**0.52 *** ± 0.01 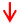	**3.13 ***^,#^ ± 0.38 
***Adcy8***	**1.0** ± 0.01	**0.43 *** ± 0.06 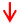	**0.43 *** ± 0.11 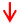
***Adcy9***	**1.0** ± 0.01	**0.85** ± 0.00	**2.70 ***^,#^ ± 0.45 
***Prkaca***	**1.0** ± 0.02	**0.77** ± 0.05	**1.70 ***^,#^ ± 0.19 
***Prkacb***	**1.0** ± 0.02	**0.96** ± 0.02	**2.43 ***^,#^ ± 0.54 
***Prkar1a***	**1.0** ± 0.01	**0.96** ± 0.02	**2.29 ***^,#^ ± 0.30 
***Prkar2a***	**1.0** ± 0.02	**0.63** ± 0.06	**1.89 ***^,#^ ± 0.20 
***Prkar2b***	**1.0** ± 0.02	**0.68** ± 0.05	**3.90 ***^,#^ ± 0.51 
***Mapk1***	**1.0** ± 0.01	**1.89 *** ± 0.09 	**3.33 ***^,#^ ± 0.65 
***Mapk3***	**1.0** ± 0.02	**0.72** ± 0.04	**1.70 ***^,#^ ± 0.29 
***Mapk6***	**1.0** ± 0.02	**1.72 *** ± 0.06 	**1.58 *** ± 0.19 
***Mapk8***	**1.0** ± 0.01	**1.56 *** ± 0.03 	**3.22 ***^,#^ ± 0.56 
***Mapk9***	**1.0** ± 0.02	**1.02** ± 0.01	**1.82 ***^,#^ ± 0.23 
***Mapk12***	**1.0** ± 0.01	**0.95** ± 0.05	**2.39 ***^,#^ ± 0.48 
***Mapk13***	**1.0** ± 0.02	**0.61** ± 0.04	**3.25 ***^,#^ ± 0.56 
***Mapk14***	**1.0** ± 0.01	**0.83** ± 0.01	**1.77 ***^,#^ ± 0.34 

## Data Availability

All relevant data and samples are available from the corresponding author on request. Further information and requests for resources and reagents should be directed to and will be fulfilled by the Lead Contact, Silvana Andric (silvana.andric@dbe.uns.ac.rs, accessed on 12 April 2021).

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
