# Peer review of "Mitochondrial Dynamics Markers and Related Signaling Molecules Are Important Regulators of Spermatozoa Number and Functionality"

_ijms, 2021, doi:10.3390/ijms22115693_

Round 1

Reviewer 1 Report

The authors provided evidence that acute and repeated stess has tremendous effects on spermatozoa through the mitochondrial activity markers. They used rats for their experiments.

It is very interesting research and deserves publication from my point of view.

Nevertheless, I believe that the authors may take into consideration and other markers that stress itself affect regarding the motility, for example markers of spermatogenesis.

Can these results extrapolate to humans?

Author Response

ijms-1200539: RESPONSES TO THE REVIEWERS

Dear Editors and Reviewers,

We greatly appreciate the opportunity to revise manuscript ijms-1200539 "Mitochondrial dynamics markers and related signaling molecules are important regulators of spermatozoa number and functionality” by Starovlah et al. Please find enclosed the revised versions (without and with “tracked changes”) for possible publication in International Journal of Molecular Sciences. 

We are very grateful to you for giving us the opportunity for minor revision and our appreciation is much more than I can express. We appreciate very much your time, patience, and help. Although “minor revision” was suggested for our manuscript and 5 days were given for revision, we kindly asked and got 3 weeks to improve our manuscript as much as it is possible. We performed new measurements, responded to all the comments and accepted all the suggestions/recommendations we could accomplish in the given time and under the pandemic situation. We deeply apologize and we are deeply sorry, but unfortunately, one recommendation, although useful, we were not able to accomplish it due to the SARS-CoV-2 pandemic and economical situation. We were not able to perform PNA test since our animal facility is temporarily closed due to construction of the new building in the university campus. Ordering will take time and the animals are not from the same origin. We apologize and we are sorry. Also, our country is importing all chemicals from the EU and/or the USA and we were not able to get chemicals. The extension will not help since the situation will not change. Since the situation will not be better in the future (just oppositely) we do not have another choice. Certainly, we are well aware of the consequences. However, all other suggestions and recommendations were accepted and performed. For assessment of spermatozoa motility (recommendation of Reviewer#1: “authors may take into consideration”), we used values of ATP and mitochondrial membrane potential, since we were not able to follow markers of spermatogenesis (recommendation of Reviewer#1).

Please see below the main changes in the manuscript (they can be tracked in red in the file named MS with tracked changes). The detailed explanation is given in the text of the file Responses to the Reviewers and every response is linked to the page (unfortunately the line numbers are not visible in the electronic version from the IJMS web site) in the version with “track changes”.

(1) The section Introduction is improved (recommendation of Reviewer#1) by adding recent publications. Please see page 3.

(2) The Research design is improved (recommendation of Reviewer#1) by using spermatozoa samples collected previously from unstressed and acutely as well as repeatedly stressed rats and performing the measurements of ATP (please see Figure 1D). Also, we used spermatozoa samples collected previously in the experiments applying different in vivo approach (stress hormone treatment in vivo) and performed the measurements of mitochondrial membrane potential (please see Supplementary Figure 1).

(3) The presentation of the Results is improved (recommendations of Reviewer#1 and Reviewer#2) by correction of the text and by adding the results of ATP values (please see Figure 1D) and mitochondrial membrane potential values (please see Supplementary Figure 1). Please see pages 3 and 4.

(4) The Figure 1 was modified by adding the results showing the ATP production (please see new Figure 1D) in spermatozoa from unstressed and acutely as well as repeatedly stressed rats. This was not the particular suggestion of the Reviewers, but Reviewer#1 recommended to asses markers of spermatogenesis for motility. Unfortunately, we were not able to perform that analysis due to the luck of the animals and chemicals. However, we had chemicals for ATP measurement and since ATP is very important marker of spermatozoa motility we performed ATP measurement. Please see page 4.

(5) The Discussion is improved (recommendations of Reviewer#2) by adding a table describing in short details the effect of IMO on the transcripts of the different pathways for a better understanding of the results. Please see pages 9-12.

(6) The Conclusions are improved (recommendations of Reviewer#1 and Reviewer#2) by correction of the text and by adding the new results. Please see page 14.

(7) Even though the reviewers did not comment, the section Materials and Methods was changed to include a description of the new methodology (ATP measurements). Please see page 14.

(8) Even though the reviewers did not comment, the section References was changed to include new references related to new information described in the Introduction and Materials and Methods. Please see pages 18-19.

(9) Even though the reviewers did not comment, the section Supplementary Material was changed and includes the new Supplemental Figure 1 (the in vivo effect of stress hormone adrenaline on the mitochondrial membrane potential of spermatozoa) as well as detailed description of the new methodology related to ATP and mitochondrial membrane potential measurements.

(10) The English language grammar and style was improved (recommendations of Reviewer#1 and Reviewer#2) with help of Mr. Strahinja Keselj (please see acknowledgment) who provides the professional peer review support to scholarly publishers, connecting journal editorial offices, authors and reviewers working in R&D-PLANet Systems Group (https://www.planetsg.com/family).

We believe that these responses have adequately addressed the questions. All items addressed by Reviewers were responded specifically and all of the suggestions were accepted. Accordingly, we believe that our results have a significant impact on human/translational physiology and we anticipate our result to be a starting point for more investigations. We believe our work will be of significant interest to investigators working in the subjects of the basic biology of reproduction and infertility, sex development disorders, and that will provide novel perspectives and approaches, as well as will stimulate the interest of the broad readership of the journal and will be of considerable interest for the diverse readership of the International Journal of Molecular Sciences in general.

Please find enclosed Supplementary Material (Supplementary Results and Methods) for all audiences as well as Supp. Results for review purpose only.

(1) All authors have made a significant contribution to the paper, have read and approved the final version of the manuscript for publication and took responsibility for the content and completeness of the manuscript and understood that if the paper, or part of the paper, is found to be faulty or fraudulent, that he/she shares responsibility with his/her coauthors.  

(2) All authors concur with the submission and that the material submitted for publication is an original work, has not been previously published whole or in part, and is not under consideration for publication elsewhere. 

(3) All authors confirm that the work will not be submitted to another journal while under consideration by the International Journal of Molecular Sciences.

(4) All authors confirm that this work is not duplicative, neither in the experiment conducted nor the text submitted.

(5) All authors declare that they have no competing financial or other interests and have nothing to disclose.

RESPONSES TO REVIEWER#1

Reviewer#1: The authors provided evidence that acute and repeated stress has tremendous effects on spermatozoa through the mitochondrial activity markers. They used rats for their experiments. It is very interesting research and deserves publication from my point of view.

RESPONSE: Thank you very much for your time, careful reading, kindness, and helpful suggestionsJ.

Reviewer#1 – Comment (1): Nevertheless, I believe that the authors may take into consideration and other markers that stress itself affect regarding the motility, for example markers of spermatogenesis.

RESPONSE1: Thank you very much for helpful suggestion. We deeply apologize and we are deeply sorry, but unfortunately, although useful, we were not able to accomplish it due to the SARS-CoV-2 pandemic and economical situation. As we stated above, our animal facility is temporarily closed due to construction of the new building in the university campus. Ordering will take time and the animals are not from the same origin. We apologize and we are sorry. Also, our country is importing all chemicals from the EU and/or the USA and we were not able to order chemicals. The extension will not help since the situation will not change. Since the situation will not be better in the future (just oppositely) we do not have another choice. Certainly, we are well aware of the consequences. However, since we had chemicals for assessment of spermatozoa motility we used values of ATP (please see new Figure1D) and mitochondrial membrane potential (new Supp. Figure 1).

Reviewer#1 – Comment (2): Can these results extrapolate to humans?

RESPONSE2: Thank you very much for the very important and useful question. Yes, results have strong translation aspect. We tried to point it in the discussion of the first version of the manuscript, but probably we did not do it properly. It is corrected in the revised version of the manuscript (page , line ). Namely, preliminary results on the small number of human samples (24) showed dramatic changes and different transcriptional profiles of mitochondrial dynamics markers in human spermatozoa of teratozoospermic (T), asthenoteratozoospermic (AT) and oligoasthenoteratozoospermic (OAT) patients compared to normozoospermic (N) patients samples (please see the PDF files provided for review purpose only). Moreover, five months ago we started collaboration with a governmental ART clinic providing the IVF service for free and according to the questionnaire completed by 89 patients, 81/89(91%) reported some degree of stress: 39/89(44%)-low degree of stress, 32/89(36%)-frequent stressful situations, 10/89(11%)-high degree of stress. We believe that these results are important since, according to the best of our knowledge, there are no published pieces of evidence considering the mitochondrial dynamics markers or their transcriptional profile as possible predictors of (in/sub)fertility. Please see page 11.

Reviewer#1 – Comments that the introduction, the research design, the presentation of the results, the conclusions and English language and style can be improved.

RESPONSES: Thank you very much for the opportunity to improve our manuscript. Please see below the changes made to improve the introduction, the research design, the presentation of the results, the conclusions and the English language and style.

(1) The Introduction is improved by adding recent publications. Please see pages 1-3.

(2) The Research design is improved by using spermatozoa samples collected previously from unstressed and acutely as well as repeatedly stressed rats and performing the measurements of ATP (please see new Figure 1D). Also, we used spermatozoa samples collected previously in the experiments applying different in vivo approach (stress hormone treatment in vivo) and performed the measurements of mitochondrial membrane potential (please see new Supplementary Figure 1).

(3) The presentation of the Results is improved by correction of the text, by adding the results of ATP values (please see new Figure 1D) and mitochondrial membrane potential values (please see new Supplementary Figure 1) as well as by adding a table describing in short details the effect of IMO on the transcripts of the different pathways for a better understanding of the results.

(4) The Conclusions are improved by correction of the text and by adding the new results.

(5) The English language, grammar and style are improved with help of Mr. Strahinja Keselj (please see acknowledgment) who provides the professional peer review support to scholarly publishers, connecting journal editorial offices, authors and reviewers working in R&D-PLANet Systems Group (https://www.planetsg.com/family).

Reviewer 2 Report

In this study authors performed two different types of psychophysical stress by immobilization (IMO), one acute the other repeated.  

Serum hormones measurement (T+DH adrenaline and corticosterone), spermatozoa isolation from caudal epididymis, along with spermatozoa concentration and functionality examinations, were carie out. Results were then correlated to  specific genes expressions after a RQ-PCR. Results showed that stress induced changes of transcriptional profile of important markers/molecules of mitochondrial dynamics/functionality.

In conclusion, stress could affect regulating both mitochondrial dynamics and spermatozoa number and functionality thus representing sort of  adaptive mechanisms.

This is an interesting study that sheds light on how the environment can change the life and functioning of cells through epigenetics-driven gene regulation.

Comments

Results

Fig. 1. The acrosome reaction is not clearly appreciable in these images. The authors should use a different technique to evidence the acrosome reaction, such as Peanut a agglutinin (PNA).

Discussion

The discussion is too long, in some places, repetitive and unclear, and should be revised if not rewritten focusing on the relevance and meaning of the results obtained and inserting this meaning into a solid background emphasizing potential effects with a relative pros and cons.

A table describing in short detail the effect of IMO on the transcripts of the different pathways should be added in the discussion for a better understanding.

Minor comments as indicated in the attached PDF.

Author Response

ijms-1200539: RESPONSES TO THE REVIEWERS

Dear Editors and Reviewers,

We greatly appreciate the opportunity to revise manuscript ijms-1200539 "Mitochondrial dynamics markers and related signaling molecules are important regulators of spermatozoa number and functionality” by Starovlah et al. Please find enclosed the revised versions (without and with “tracked changes”) for possible publication in International Journal of Molecular Sciences. 

We are very grateful to you for giving us the opportunity for minor revision and our appreciation is much more than I can express. We appreciate very much your time, patience, and help. Although “minor revision” was suggested for our manuscript and 5 days were given for revision, we kindly asked and got 3 weeks to improve our manuscript as much as it is possible. We performed new measurements, responded to all the comments and accepted all the suggestions/recommendations we could accomplish in the given time and under the pandemic situation. We deeply apologize and we are deeply sorry, but unfortunately, one recommendation, although useful, we were not able to accomplish it due to the SARS-CoV-2 pandemic and economical situation. We were not able to perform PNA test since our animal facility is temporarily closed due to construction of the new building in the university campus. Ordering will take time and the animals are not from the same origin. We apologize and we are sorry. Also, our country is importing all chemicals from the EU and/or the USA and we were not able to get chemicals. The extension will not help since the situation will not change. Since the situation will not be better in the future (just oppositely) we do not have another choice. Certainly, we are well aware of the consequences. However, all other suggestions and recommendations were accepted and performed. For assessment of spermatozoa motility (recommendation of Reviewer#1: “authors may take into consideration”), we used values of ATP and mitochondrial membrane potential, since we were not able to follow markers of spermatogenesis (recommendation of Reviewer#1).

Please see below the main changes in the manuscript (they can be tracked in red in the file named MS with tracked changes). The detailed explanation is given in the text of the file Responses to the Reviewers and every response is linked to the page (unfortunately the line numbers are not visible in the electronic version from the IJMS web site) in the version with “track changes”.

(1) The section Introduction is improved (recommendation of Reviewer#1) by adding recent publications. Please see page 3.

(2) The Research design is improved (recommendation of Reviewer#1) by using spermatozoa samples collected previously from unstressed and acutely as well as repeatedly stressed rats and performing the measurements of ATP (please see Figure 1D). Also, we used spermatozoa samples collected previously in the experiments applying different in vivo approach (stress hormone treatment in vivo) and performed the measurements of mitochondrial membrane potential (please see Supplementary Figure 1).

(3) The presentation of the Results is improved (recommendations of Reviewer#1 and Reviewer#2) by correction of the text and by adding the results of ATP values (please see Figure 1D) and mitochondrial membrane potential values (please see Supplementary Figure 1). Please see pages 3 and 4.

(4) The Figure 1 was modified by adding the results showing the ATP production (please see new Figure 1D) in spermatozoa from unstressed and acutely as well as repeatedly stressed rats. This was not the particular suggestion of the Reviewers, but Reviewer#1 recommended to asses markers of spermatogenesis for motility. Unfortunately, we were not able to perform that analysis due to the luck of the animals and chemicals. However, we had chemicals for ATP measurement and since ATP is very important marker of spermatozoa motility we performed ATP measurement. Please see page 4.

(5) The Discussion is improved (recommendations of Reviewer#2) by adding a table describing in short details the effect of IMO on the transcripts of the different pathways for a better understanding of the results. Please see pages 9-12.

(6) The Conclusions are improved (recommendations of Reviewer#1 and Reviewer#2) by correction of the text and by adding the new results. Please see page 14.

(7) Even though the reviewers did not comment, the section Materials and Methods was changed to include a description of the new methodology (ATP measurements). Please see page 14.

(8) Even though the reviewers did not comment, the section References was changed to include new references related to new information described in the Introduction and Materials and Methods. Please see pages 18-19.

(9) Even though the reviewers did not comment, the section Supplementary Material was changed and includes the new Supplemental Figure 1 (the in vivo effect of stress hormone adrenaline on the mitochondrial membrane potential of spermatozoa) as well as detailed description of the new methodology related to ATP and mitochondrial membrane potential measurements.

(10) The English language grammar and style was improved (recommendations of Reviewer#1 and Reviewer#2) with help of Mr. Strahinja Keselj (please see acknowledgment) who provides the professional peer review support to scholarly publishers, connecting journal editorial offices, authors and reviewers working in R&D-PLANet Systems Group (https://www.planetsg.com/family).

We believe that these responses have adequately addressed the questions. All items addressed by Reviewers were responded specifically and all of the suggestions were accepted. Accordingly, we believe that our results have a significant impact on human/translational physiology and we anticipate our result to be a starting point for more investigations. We believe our work will be of significant interest to investigators working in the subjects of the basic biology of reproduction and infertility, sex development disorders, and that will provide novel perspectives and approaches, as well as will stimulate the interest of the broad readership of the journal and will be of considerable interest for the diverse readership of the International Journal of Molecular Sciences in general.

Please find enclosed Supplementary Material (Supplementary Results and Methods) for all audiences as well as Supp. Results for review purpose only.

(1) All authors have made a significant contribution to the paper, have read and approved the final version of the manuscript for publication and took responsibility for the content and completeness of the manuscript and understood that if the paper, or part of the paper, is found to be faulty or fraudulent, that he/she shares responsibility with his/her coauthors.  

(2) All authors concur with the submission and that the material submitted for publication is an original work, has not been previously published whole or in part, and is not under consideration for publication elsewhere. 

(3) All authors confirm that the work will not be submitted to another journal while under consideration by the International Journal of Molecular Sciences.

(4) All authors confirm that this work is not duplicative, neither in the experiment conducted nor the text submitted.

(5) All authors declare that they have no competing financial or other interests and have nothing to disclose.

RESPONSES TO REVIEWER#2

Referee#2: In this study authors performed two different types of psychophysical stress by immobilization (IMO), one acute the other repeated.  Serum hormones measurement (T+DH adrenaline and corticosterone), spermatozoa isolation from caudal epididymis, along with spermatozoa concentration and functionality examinations, were carried out. Results were then correlated to specific genes expressions after a RQ-PCR. Results showed that stress induced changes of transcriptional profile of important markers/molecules of mitochondrial dynamics/functionality. In conclusion, stress could affect regulating both mitochondrial dynamics and spermatozoa number and functionality thus representing sort of adaptive mechanisms. This is an interesting study that sheds light on how the environment can change the life and functioning of cells through epigenetics-driven gene regulation.

 RESPONSE: We are really very grateful and we appreciate very much your time, careful reading, patience, kindness and helpful suggestions. Our appreciation is much more than we can express J.

Reviewer#2 – Comment (1): Results - Fig. 1. The acrosome reaction is not clearly appreciable in these images. The authors should use a different technique to evidence the acrosome reaction, such as Peanut a agglutinin (PNA).

RESPONSE1: Thank you very much for very useful suggestion. We agree that it will be worthy to use PNA technique. However, as we stated above, conditions did not permit. We deeply apologize and we are deeply sorry, but unfortunately, although your suggestion is very useful, we were not able to accomplish it due to the SARS-CoV-2 pandemic and economical situation. We were not able to perform PNA test since our animal facility is temporarily closed due to construction of the new building in the university campus. Ordering will take time and the animals are not from the same origin. We apologize and we are sorry. Also, our country is importing all chemicals from the EU and/or the USA and we were not able to order chemicals. The extension will not help since the situation will not change. Since the situation will not be better in the future (just oppositely) we do not have another choice. Certainly, we are well aware of the consequences. However, all other suggestions and recommendations were accepted and performed and some other measurements were performed with samples we collected previously. Please see responses above.

Reviewer#2 – Comment (2): Discussion - The discussion is too long, in some places, repetitive and unclear, and should be revised if not rewritten focusing on the relevance and meaning of the results obtained and inserting this meaning into a solid background emphasizing potential effects with a relative pros and cons.

RESPONSE2: Thank you very much for helpful suggestion. We did our best to modify discussion by shortening, deleting the repetitive and unclear text and to rewriting it. However, there are not much published pieces of evidence on the subject of the manuscript. Also, we wanted to add some translational aspect of the results.

Reviewer#2 – Comment (3): Discussion - A table describing in short detail the effect of IMO on the transcripts of the different pathways should be added in the discussion for a better understanding.

RESPONSE3: Thank you very much for very helpful suggestion. The table describing in short detail the effect of IMO on the transcripts of elements of the different pathways is added in the discussion for a better understanding of the results.

Reviewer#2 – Comments that the presentation of the results, the conclusions and English language and style can be improved.

RESPONSES: Thank you very much for very much for the opportunity to improve our manuscript. Please see below the changes made to improve the presentation of the results, the conclusions and the English language and style.

(1) The presentation of the Results is improved by correction of the text, by adding the results of ATP values (please see new Figure 1D) and mitochondrial membrane potential values (please see new Supplementary Figure 1) as well as by adding a table describing in short details the effect of IMO on the transcripts of the different pathways for a better understanding of the results.

(2) The Conclusions are improved by correction of the text and by adding the new results.

(3) The English language, grammar and style are improved with help of Mr. Strahinja Keselj (please see acknowledgment) who provides the professional peer review support to scholarly publishers, connecting journal editorial offices, authors and reviewers working in R&D-PLANet Systems Group (https://www.planetsg.com/family).

RESPONSES ON MINOR COMMENTS NOTED IN THE PDF FILE REVIWER KINDLY PROVIDED

Reviewer 2 – Minor Comment (1): (line 77) explain the abbreviation in full.

MINOR RESPONSE1: Thank you very much for very helpful suggestion. We apologize and we are deeply sorry for the obvious mistake. Abbreviation is explained in full.

Reviewer 2 – Minor Comment (2): (line 78) explain the abbreviation in full.

MINOR RESPONSE2: Thank you very much for very useful suggestion. We apologize and we are deeply sorry for the obvious mistake. Abbreviation is explained in full.

Reviewer 2 – Minor Comment (3): (line 91) check the grammar.

MINOR RESPONSE3: Thank you very much for very helpful suggestion. We apologize and we are deeply sorry for the obvious mistake. As it was stated above, the English language, grammar and style are improved with help of Mr. Strahinja Keselj (please see acknowledgment) who provides the professional peer review support to scholarly publishers, connecting journal editorial offices, authors and reviewers working in R&D-PLANet Systems Group (https://www.planetsg.com/family).

Reviewer 2 – Minor Comment (4): (lines 95-96) explain the abbreviation in full.

MINOR RESPONSE4: Thank you very much for very useful suggestion. We apologize and we are deeply sorry. Abbreviation is explained in full.

Reviewer 2 – Minor Comment (5): (lines 98-100) check the abbreviations: if used for the first time they should be explained in full.

MINOR RESPONSE5: Thank you very much for very helpful suggestion. We apologize and we are deeply sorry. Abbreviations are explained in full.

Reviewer 2 – Minor Comment (6): (Figure 1, lines 139-140) Acrosome reaction is not clearly appreciable in these images. Probably, the authors should have chosen a different technique to evidence the acrosome reaction, such as Peanut a agglutinin (PNA).

MINOR RESPONSE6: Thank you very much for very useful suggestion. As it is stated above, we completely agree that it will be worthy to use PNA technique. However, as it is stated above, conditions did not permit. We deeply apologize and we are deeply sorry, but unfortunately, although your suggestion is very useful, we were not able to accomplish it due to the SARS-CoV-2 pandemic and economical situation. We were not able to perform PNA test since our animal facility is temporarily closed due to construction of the new building in the university campus. Ordering will take time and the animals are not from the same origin. We apologize and we are sorry. Also, our country is importing all chemicals from the EU and/or the USA and we were not able to order chemicals. The extension will not help since the situation will not change. Since the situation will not be better in the future (just oppositely) we do not have another choice. Certainly, we are well aware of the consequences. However, all other suggestions and recommendations were accepted and performed and some other measurements were performed with samples we collected previously. Please see responses above.

Reviewer 2 – Minor Comment (7): (line 245) For the first time, our results.

MINOR RESPONSE7: Thank you very much for very helpful suggestion. We apologize and we are deeply sorry for the mistake. The sentence is corrected.

Reviewer 2 – Minor Comment (8): (line 249) This phrase can be deleted.

MINOR RESPONSE8: Thank you very much for very useful suggestion. We apologize and we are deeply sorry for the mistake. The phrase was deleted.

Reviewer 2 – Minor Comment (9): (lines 250-254) Not clear. Simplify and reduce.

MINOR RESPONSE9: Thank you very much for very useful suggestion. We apologize and we are deeply sorry for the mistake. The text was corrected.

Reviewer 2 – Minor Comment (10): (lines 254-257) Redundant and repetitive.

MINOR RESPONSE10: Thank you very much for very helpful suggestion. We apologize and we are deeply sorry for the mistake. The redundant and repetitive text was removed. Please see the manuscript with “tracked changes”.

Reviewer 2 – Minor Comment (11): (lines 250-257) Please, be more concise and not repetitive.

MINOR RESPONSE11: Thank you very much for very helpful suggestion. We apologize and we are deeply sorry for the mistake. The repetitive text was removed. Please see the manuscript with “tracked changes”.

Reviewer 2 – Minor Comment (12): (lines 289-296) Repetition of introduction and results.

MINOR RESPONSE12: Thank you very much for very useful suggestion. We apologize and we are deeply sorry for the mistake. The repetitive text was removed. Please see the manuscript with “tracked changes”.

Reviewer 2 – Minor Comment (13): (lines 300-302) Check English grammar.

MINOR RESPONSE13: Thank you very much for very helpful suggestion. We apologize and we are deeply sorry for the obvious mistake. As it was stated above, the English language, grammar and style are improved with help of Mr. Strahinja Keselj (please see acknowledgment) who provides the professional peer review support to scholarly publishers, connecting journal editorial offices, authors and reviewers working in R&D-PLANet Systems Group (https://www.planetsg.com/family).

Reviewer 2 – Minor Comment (14): (lines 302-308) Not clear at all.

MINOR RESPONSE14: Thank you very much for very useful suggestion. We apologize and we are deeply sorry for the mistake. We did our best to make text more clear. Please see the manuscript with “tracked changes”.

Reviewer 2 – Minor Comment (15): (line 312) Why??? not clear.

MINOR RESPONSE15: Thank you very much for very helpful suggestion. We apologize and we are deeply sorry for the mistake. We did our best to make the text more clear. Please see the manuscript with “tracked changes”.

Reviewer 2 – Minor Comment (16): (line 316) Why? rat, human, obese confusing.

MINOR RESPONSE16: Thank you very much for very useful suggestion. We apologize and we are deeply sorry for the mistake. We did our best to make the text more clear. Please see the manuscript with “tracked changes”.

Reviewer 2 – Minor Comment (17): (lines 316-318) No data reported.

MINOR RESPONSE17: Thank you very much for very helpful suggestion. The journal article that was cited after the sentence concluded that perturbed mitochondrial release of Cyt C may be the distinguishing molecular feature between oligospermia and asthenospermia.

Reviewer 2 – Minor Comment (18): (lines 323-325) Repetitive.

MINOR RESPONSE18: Thank you very much for very useful suggestion. We apologize and we are deeply sorry for the mistake. The repetitive text was removed. Please see the manuscript with “tracked changes”.

Reviewer 2 – Minor Comment (19): (lines 330-333) Avoid this phrases.

MINOR RESPONSE19: Thank you very much for very helpful suggestion. We apologize and we are deeply sorry for the mistake. The phrases were removed. Please see the manuscript with “tracked changes”.

Reviewer 2 – Minor Comment (20): (lines 330-333) Repetition of lines 318-322.

MINOR RESPONSE20: Thank you very much for very useful suggestion. We apologize and we are deeply sorry for the mistake. The repetitive text was removed. Please see the manuscript with “tracked changes”.

Reviewer 2 – Minor Comment (21): (lines 350-380) Too long, repetitive redundant, not linked to the experiments of the study.

MINOR RESPONSE21: Thank you very much for the suggestion. We apologize and we are deeply sorry for not being able to agree with you although we strongly would like. Namely, this part of the text is related to the translational aspects of the study and describes our results obtained using the small amount of human subjects. Also, Review#1 asked to comment translational aspect (please see attached PDF with responses to both Reviewers). We apologize, but the approach we used is new and there are not many published pieces of evidence, but there are some relations with other studies and that is why we kept the references.
